# Impact of Cattle Feeding Strategy on the Beef Metabolome

**DOI:** 10.3390/metabo12070640

**Published:** 2022-07-13

**Authors:** Juan Fernando Morales Gómez, Nara Regina Brandão Cônsolo, Daniel Silva Antonelo, Mariane Beline, Mohammed Gagaoua, Angel Higuera-Padilla, Luiz Alberto Colnago, David Edwin Gerrard, Saulo Luz Silva

**Affiliations:** 1Department of Animal Science, College of Animal Science and Food Engineering, University of São Paulo, Pirassununga 13635-900, SP, Brazil; juanmoralesg14@gmail.com (J.F.M.G.); nara.consolo@usp.br (N.R.B.C.); marianebeline@gmail.com (M.B.); 2Department of Animal Nutrition and Production, College of Veterinary Medicine and Animal Science, University of São Paulo, Pirassununga 13635-900, SP, Brazil; danielantonelo@usp.br; 3Food Quality and Sensory Science Department, Teagasc Food Research Centre, Ashtown, D15 KN3K Dublin, Ireland; mohammed.gagaoua@teagasc.ie; 4EMBRAPA Instrumentação, XV de Novembro 1452, São Carlos 13560-970, SP, Brazil; angelruben@hotmail.com (A.H.-P.); luiz.colnago@embrapa.br (L.A.C.); 5School of Animal Sciences, Virginia Tech, Blacksburg, VA 24061, USA; dgerrard@vt.edu

**Keywords:** feeding system, growth rate, metabolomics, Nellore

## Abstract

The present study explored changes in the meat metabolome of animals subjected to different finishing systems and growth rates. Thirty-six Angus × Nellore crossbred steers were used in a completely randomized design with four treatments: (1) feedlot system with high average daily gain (ADG; FH); (2) feedlot system with low ADG (FL); (3) pasture system with high ADG (PH); and (4) pasture system with low ADG (PL). After harvest and chilling, *Longissimus thoracis* (LT) muscle samples were taken for metabolite profile analysis using nuclear magnetic resonance. Spectrum was analyzed using chenomx software, and multi- and mega-variate data analyses were performed. The PLS-DA showed clear separation between FH and PL groups and overlap among treatments with different finishing systems but similar for matching ADG (FL and PH) treatments. Using a VIP cut-off of around 1.0, ATP and fumarate were shown to be greater in meat from PL cattle, while succinate, leucine, AMP, glutamate, carnosine, inosine, methionine, G1P, and choline were greater in meat from FH. Comparing FL and PH treatments, glutamine, carnosine, urea, NAD+, malonate, lactate, isoleucine, and alanine were greater in the meat of PH cattle, while G6P and betaine were elevated in that of FL cattle. Relevant pathways were also identified by differences in growth rate (FH versus PL) and finishing system were also noted. Growth rate caused a clear difference in meat metabolism that was highlighted by energy metabolism and associated pathways, while the feeding system tended to alter protein and lipid metabolism.

## 1. Introduction

Feedlot finishing systems are typically based on high-energy diets meticulously balanced to provide all essential nutrients necessary to support maximal tissue growth. As a result, this approach to feeding cattle may alter muscle metabolism to support greater average daily gains (ADG) and greater fat deposition, all leading to greater performance in feedlot cattle compared to animals finished on pasture [1]. On the other hand, muscle from animals finished on pasture must adapt to lower dietary nutrient densities and greater physical activity, which can also shift metabolism to a more oxidative type, thereby slowing tissue accretion rates compared to feedlot animals [1,2,3,4].

Many studies have reported differences in meat quality characteristics that arise from changes in beef cattle growth rate, especially regarding the vast differences created between grass- and grain-fed production systems. This is not particularly surprising given changes in nutritional status and(or) environmental stimuli alter gene expression and elicit a whole host of changes in tissue metabolic pathways and subsequent metabolite abundances [4,5,6,7,8,9]. Even so, only a few studies have explored possible mechanisms undergirding the transcriptome and metabolome, namely in the rumen [10], spleen [11], muscle [12], and, more recently, in the liver and blood [13]. However, the molecular mechanisms responsible for inducing changes in meat quality parameters between these two highly competing production paradigms, where diet and growth rate can be quite diverse, have not been fully investigated.

Metabolomics is a powerful foodomics tool that, in recent years, has allowed for broad exploration of skeletal muscle metabolism in farm animals [12,14,15,16]. Yet, there is currently still a dearth of information regarding the usefulness of metabolomics to better understand the changes occurring in the muscles of cattle produced in different feeding systems. Therefore, this study aimed to explore, for the first time, changes in the meat metabolome of cattle subjected to a wide array of production scenarios spanning the diversity of beef production across the globe.

## 2. Results 

Based on the ^1^D H^1^-NMR analyses, 37 compounds were identified in the meat extracts, including essential and non-essential amino acids, metabolites belonging to glycolysis, the citric acid cycle, metabolism of purines, fatty acids, nitrogen, and others. The descriptive statistics are given in Table 1.

The PLS-DA score and loading plot analysis allowed for visualizing the differences in the metabolite profiles among the treatments (Figure 1A,B). A clear separation between FH and PL groups were noted, but similarities among treatments with different finishing system and similar ADG (FL and PH) can be observed, albeit with more overlap (Figure 1A).

A SAM plot displayed a linear correlation between the expected and observed values determined for the quantified metabolites (Figure 2) for the pairwise comparisons: FH versus PL (Figure 2A) and FL versus PH (Figure 2B). A total of four compounds were identified that differentiate between FH and PL beef, namely leucine, fumarate, ATP, and succinate (Table 2). Seven compounds differentiated FL from PH beef and included G1P, carnosine, glutamine, urea, NAD+, isoleucine, and malonate (Table 2).

The VIP analysis based on the PLS-DA shows the top 15 metabolites that changed in relative amounts between treatments. The y-axis represents the molecules in order of importance for group classification from top to bottom. Using a VIP cut-off of around 1.0, ATP and fumarate were greater in beef from PL cattle, while succinate, leucine, AMP, glutamate, carnosine, inosine, methionine, G1P, and choline were greater in that of FH cattle (Figure 3A). Comparing FL and PH cattle, glutamine, carnosine, urea, NAD+, malonate, lactate, isoleucine, and alanine were elevated in PH beef, while G6P and betaine were elevated in the meat of FL animals (Figure 3B). 

Relevant pathways identified by differences in growth rate (FH versus PL) were correlated with energy and protein metabolism, which included alanine, aspartate, and glutamate metabolism, aminoacyl-tRNA biosynthesis, arginine biosynthesis, butanoate metabolism, histidine metabolism, citrate cycle (TCA cycle), galactose metabolism, D-glutamine, and D-glutamate metabolism and nitrogen metabolism (Figure 4A and Table 3). When pathway analysis was performed to evaluate differences in feeding system (FL versus PH), aminoacyl-tRNA biosynthesis, arginine biosynthesis, alanine, aspartate and glutamate metabolism, neomycin, kanamycin and gentamicin biosynthesis, D-glutamine and D-glutamate metabolism, nitrogen metabolism and valine, and leucine and isoleucine biosynthesis were the main pathways influenced (Figure 4B and Table 3).

## 3. Discussion

This study provides a comprehensive analysis of changes in metabolites and metabolic pathways in meat from beef cattle fed one of two feeding regimes at one of two different growth rates. The totality of these results shows the complexity of metabolic regulation possible in producing this highly coveted food source over a broad range of production scenarios and argues the necessity of understanding the molecular mechanisms driving meat quality characteristics in animal production and meat science. The present metabolomic data are supported by meat quality data previously described by Gómez et al. [7]. The pairwise comparison between FH and PL animals highlights several differences in the performance, carcass, and meat quality traits [7]. According to the results of this study, metabolic profiles differ greatly between these two groups. These changes mostly reflect differences in growth rate because these extremes (FH versus PL) represent the highest and lowest average daily gains possible.

It is well-known that growth is a complex process controlled by many genes, environmental factors, and epigenetic and pleiotropic mechanisms [17]. Data presented herein support this assertion but begin to help us unravel the biology driving improved performance in growing cattle over a wide range of production scenarios. Differences in metabolite concentrations between these diverse groups represented nine metabolic pathways, which are related to the three general pathways involving protein, lipid, and carbohydrate metabolisms.

Higher concentrations of free amino acids, such as leucine, glutamate, and methionine in meat from FH animals compared to that of PL cattle are likely related to changes in muscle protein metabolism, where faster-growing livestock, in this case FH cattle, generally produce more glycolytic muscle, while slower growing (PL) animals likely have more oxidative muscle [1,2,7]. The relative importance of ‘muscle type’ in facilitating greater performance was initially proposed from studies using wild and domestic pigs [18], and anecdotally by the massive increases in the glycolytic nature of muscle in fast-growing, modern-day broiler lines [19]. Wild pigs contain muscle with a higher proportion of red, oxidative, slow-contracting muscle fibers. Curiously, these animals have greatly diminished muscle growth potential and only about half the number of fast fibers (IIB), suggesting muscle growth and subsequent performance may be related to the type of muscle produced. Karlsson et al. [20] supported this hypothesis by showing selection within domesticated pigs for growth resulted in an increase in the relative proportion of IIB fibers. This paradigm is not restricted to changes in response to selection because pigs, rodents, sheep, and cattle fed on various commercially available beta-adrenergic agonists (growth promotants) that stimulate whole body growth rate also produce faster-contracting, more glycolytic muscle than non-treated controls [21,22,23]. While the aforementioned fail to establish a direct cause and effect relationship between growth rate and muscle type, we recently showed that mice lacking a functional IIB gene and thereby blunt expression of the fastest muscle fiber phenotype in adult muscle are refractile to known genetic and chemical approaches that typically augment growth [24]. As such, changes in tissue amino acids reported herein are likely related to changes in protein turnover, as impacted by changes in muscle type composition. Greater amounts of free amino acids found in the meat from FH cattle argue greater amounts of muscle protein accretion occur in faster-growing cattle. Increases in protein deposition result from net differences between protein synthesis and degradation in muscle. While not completely intuitive, growth rate results in increases in both protein synthesis and degradation [25]. As such, a period of increased growth rate prior to harvest is preferred from a beef quality standpoint because faster-growing cattle produce more tender beef [26]. Consistent with this line of reasoning, Antonelo et al. [8] found that muscle structure-related proteins were in greater abundance in the meat of PL versus FH cattle suggesting greater proteolysis occurred in muscle of FH cattle. This is also consistent with tenderness data of these same animals, as reported by Gómez et al. [7], where meat from FH cattle was more tender compared to that from PL cattle. Delineating the exact process by which this occurs will require additional studies focused on protein synthesis and degradation processes occurring during feeding cattle. 

Notwithstanding, Imaz et al. [27] evaluated blood metabolomics related to cattle growth rate and observed a negative relationship between growth and blood concentrations of valine, isoleucine, leucine, pyruvate, and acetyl groups. Authors speculated that cattle with faster growth rates clear these metabolites more rapidly from circulation, facilitating greater tissue growth. This possibility is supported by the fact that we observed higher amounts of amino acids in the muscle of faster-growing cattle. Along this same line of reasoning, Widmann et al. [17] showed that growth-related changes in metabolisms at the onset of puberty in heifers elicited a torrent of processes in cells and showed that arginine metabolism is highly correlated with several key cellular pathways in these processes. Therefore, it is unsurprising that arginine biosynthesis is one of the most important pathways observed in enrichment analyses differentiating growth rate in cattle in our study. Even so, direct comparisons of the results from this study and others are somewhat difficult because of differences in physiological age, gender, and sample evaluation. 

Choline was higher in meat from FH cattle than PL animals. This metabolite plays an important role in phospholipid synthesis [28], possibly as a result of greater concentrations of intramuscular lipids in FH animals [7]. Imaz et al. [27] explained that high-performing cattle are characterized by a more rapid uptake of metabolites such as acetyl groups and 3-hydroxybutyrate from blood, thereby increasing lipid synthesis. Both compounds are indeed greater in the circulation of faster-growing animals [27]. Changes in growth rate are closely related to changes in muscle fiber type composition and their collective metabolisms [29]. Data presented herein show that most of the metabolites from carbohydrate metabolism are correlated with glycogenolysis and the triacyl carboxylic acid (TCA) cycle. Increased amounts of succinate, AMP, and G1P in the meat of FH animals may be partially explained by the fact that the muscle of FH animals is more glycolytic in nature and uses substrates and pathways that support this type of metabolism. In this regard, the ability of the tissue to store and mobilize carbohydrate would result in greater amounts G1P, perhaps to support glycolysis and the TCA cycle compared to that of slower growing cattle. This increased capacity to mobilize this type of energy substrates may make FH animals more efficient in supporting greater growth rates, or vice versa. In support of this notion, Costa et al. [30] reported blood metabolites were highly correlated with glycolysis and TCA pathways in animals with a high potential for post-weaning growth. The authors postulated that greater growth requires greater energy production through carbohydrate-specific pathways [30], again arguing a relationship between fast-growing animals and carbohydrate metabolism in muscle. The ability to store, produce, and recycle energy via creatine phosphate has also been reported as an important metabolic pathway for animals with greater live performance [27].

Differences in muscle adenonucleotide data reported herein further support our contention that muscle type changes with growth rate, as facilitated by diet. Basal energy (ATP) charge between oxidative and glycolytic muscle differs, as do resting phosphocreatine levels, which is an immediate source of high-energy phosphates during the rapid loss of ATP. Muscle samples collected from these cattle occurred at 24 h postmortem. However, completion of rigor mortis in beef carcasses varies [31]. It could be impacted by muscle type, either as driven by the predominant form of energy metabolism or temperature decline during the postmortem period. Regrettably, we did not monitor temperature declines across treatments, but differences likely occurred given differences in carcass weights and(or) fatness among treatments. Greater amounts of ATP may have been detected in the muscle of PL cattle because of delayed completion of rigor, defined as a complete loss of ATP production rather than loss of extensibility and the overall oxidative nature of the tissue. The reason for the greater amount of AMP in the muscle of FH cattle, is also related to the energy charge of the muscle, and may be related to AMP deaminase amount or activity [32], which varies with muscle type [33], though the ultimate pH of meat remained the same [7,34]. In agreement with these metabolomics data, Antonelo et al. [8] also showed that FH animals possess an over-abundance of enzymes related to glycolysis/gluconeogenesis pathways, suggesting greater glycolytic metabolism in fed cattle. Finally, muscles with more glycolytic metabolism tend to have more tender meat and a brighter appearance than more oxidative muscle [2,29], which agrees with the meat quality characteristics reported by Gómez et al. [7]. 

Regarding the pairwise comparison between FL and PH cattle, similarities in animal performance and carcass traits were reported by Gómez et al. [7], so comparing these treatments may allow for a better understanding of the broad differences between pasture and feedlot feeding systems. Many studies have reported the differences in performance and meat quality of cattle finished in different production systems, grass or grain, including changes in muscle fiber type composition, intramuscular fat accumulation, meat tenderness, and color [1,2,4,5,6,35]. However, the molecular mechanism and pathways responsible for differences between grass and grain-fed animals are still largely unknown. 

The overlapping nature of the metabolic profiles between FL and PH treated cattle suggests the similarities in their meat metabolites profiles are related to similar performance parameters, even though some differences in meat quality have been observed between these two groups [7]. However, the VIP score suggested differences in the metabolite profiles that may be explained by differences in oxidative, protein, and lipid metabolism.

Because pastures are rich in antioxidant compounds, it is unsurprising that meat from grass-fed animals possesses large amounts of these compounds. Consistent with this theme, meat from grass-fed animals has more glutamine and carnosine, two metabolites that play a versatile role in cell metabolism, including cellular redox reactions and antioxidative capacities [36]. Greater antioxidant capacity in the muscle of grass-fed animals might lead to more effective defenses against muscle cell apoptosis during the early postmortem period compared to grain-fed animals with similar growth rates. These changes in cellular function could contribute to differences in meat tenderness and maturation processes between PH and FL animals and result in differences in lean color, as reported by Gómez et al. [7]. These data are further supported by Antonelo et al. [8], who found a greater abundance of Annexin 2 (ANXA2) in PH animals than in FL cattle. ANXA2 has anti-apoptotic properties that protect cells from damage [37] and is positively correlated with WBSF [38].

Differences in protein metabolism are also suggested by the observation that greater urea and amino acids are detected in the meat of PH cattle. While this suggests greater protein turnover in the muscle of PH cattle, it may also be related to the catabolism of amino acids for a carbon source, hence greater deamination. Animals fed pasture-based diets most likely consume less fermentable carbohydrates compared to FL cattle. In this regard, PH cattle probably produced less propionate than feedlot-managed cattle, which is more glucogenic than acetate. These changes mirror muscle fiber type in pasture-fed that moved to a more oxidative type of metabolism than cattle in feedlot systems [7]. This is also consistent with our observation that greater amounts of G6P were detected in the muscle of FL cattle, which reflects more glucose availability in the muscle tissue, reflecting the differences in dietary intake, where animals finished in the feedlot had greater soluble carbohydrate intakes and consequently greater liver gluconeogenesis. Similarly, Jia et al. [13], studying blood and liver transcriptomic and metabolomic of grass and grain-fed cattle, reported that grass feeding induces upregulation of those genes involved in fatty acid degradation and amino acid metabolism (catabolism) to meet the energy demands of the body. Carrillo et al. [12] also reported that feeding changes specific gene expressions to support greater glucose metabolism, which is greater in the muscle of intensively fed cattle.

Finally, an interesting metabolite differing between the tissue of FL and PH cattle was malonate. Malonate is a key compound in malonyl-CoA synthesis and represents the rate-limiting and committed metabolite in de novo fatty acid synthesis [39]. The lower concentration of this compound in FL cattle may indicate a greater demand for fatty acids synthesis, which agrees with greater intramuscular fat in FL compared to PH cattle [7]. Consistent with this line of reasoning, Jia et al. [13] found increases in liver gene expression related to glycolysis/gluconeogenesis, fatty acids degradation, and amino acid metabolism pathways in grass-fed animals, presumably to meet their differing energy demand. Conversely, other studies report the up-regulation of lipogenic enzymes in tissues of heavily fed cattle to that from cattle fed grass diets [9,40].

## 4. Material and Methods

### 4.1. Animals and Treatments

The present data is part of a larger experiment partially published by Gómez et al. [7]. Thirty-six Angus × Nellore crossbred steers (330 ± 30 kg body weight (BW), 12 ± 1 months old) were used in a completely randomized design with 4 treatments: (1) feedlot system with high average daily gain (FH; ADG estimated to be 1.5 kg/d); (2) feedlot system with low ADG (FL; ADG estimated to be 0.9 kg/d); (3) pasture system with high ADG (PH; ADG estimated to be 0.9 kg/d); and (4) pasture system with low ADG (PL; ADG estimated to be 0.6 kg/d). Animals finished in the feedlot system were fed a diet consisting of 80% concentrate and 20% roughage. Animals FH were fed *ad libitum* while FL animals were fed 70% of ad libitum diets. Animals finished in the pasture system were housed in paddocks consisting of Marandu grass (Brachiaria brizantha cv. Marandu), and the ADG was controlled by stocking density. Moreover, animals were harvested based on a final body weight (530 ± 20 kg BW). Animals reached a final weight of 530 kg after 116, 228, 262 and 292 d of feeding with an average daily gain of 1.50, 0.94, 0.75 and 0.62 kg/ in FH, FL, PH and PL groups, respectively. More details on rearing practices, animals performance, carcass, and meat quality traits are described in Gómez et al. [7].

### 4.2. Harvest and Sample Collection

When each treatment reached the target BW, animals were transported to an abattoir located at the University of Sao Paulo (1 km and 4 km for feedlot and pasture-fed animals, respectively), and harvested according to the Sanitary and Industrial Inspection Regulation for Animal Origin Products of Humanitarian Slaughter Guidelines as required by Brazilian law. All procedures were similar to those used in commercial settings. After 24 h of chilling (0 to 2 °C), carcasses were fabricated, and *Longissimus thoracis* (LT) muscle samples (approximately 10 g) were taken from twenty-four animals, six per treatment, adjacent to the 12th rib and stored at −80 °C for further analysis and metabolite profiles.

### 4.3. Extraction of Polar Metabolites from Meat

Six samples per group were randomly selected, and approximately 0.5 g of meat frozen (*n* = 24, 6 per group) was macerated and homogenized using a blender (Ultra-Turrax®, T 25 digital, IKA, Campinas, SP, Brazil). Metabolites were extracted with 3.5 mL of cold methanol/chloroform/water solution (2:2:1 v/v) while vortexing for 1 min, as previously described [41]. Samples were subsequently placed on ice for 15 min and were then centrifuged for 15 min at 10,000× *g* at 4 °C to remove precipitated protein and connective tissue. Supernatants were carefully transferred to Eppendorf tubes and freeze-dried (Itasul Import and Instrumental Technical Ltda, Porto Alegre, RS, Brazil). The freeze-dried residue was reconstituted in 600 μL 100 mM phosphate buffer (containing 10% D_2_O and 90% H_2_0, pH 7.0) and 60 μL internal standard solution (containing 5 mM 3-[trimethylsilyl]-1-propanesulfonic acid sodium salt [DSS]) as a quantitation standard and chemical shift reference, and 100 mM imidazole as a pH indicator was added. Samples were centrifuged at 10,000× *g* for 3 min at 4 °C to remove any precipitate. For the analysis, 600 μL of the supernatant was transferred to standard 5 × 178 mm thin-walled NMR tubes (VWR International).

### 4.4. Extraction of Polar Metabolites from Meat

^1^H-NMR was used for meat metabolite profiling, and the analyses were performed at EMBRAPA Instrumentação (São Carlos, SP, Brazil) following the methodology described by Cônsolo et al. [14]. Briefly, ¹H NMR spectra were acquired at 300 K on a Bruker Avance 14.1 T spectrometer (Bruker Corporation, Karlsruhe, Baden-Württemberg, Germany) at 600.13 MHz for 1H, using a BBO 5 mm probe. D2O was used as a lock solvent and DSS as the chemical shift reference for 1H. Standard one-dimensional (1D) proton NMR spectra were acquired using a single 90° pulse experiment, and each spectrum was the sum of 64 FIDs. Water suppression was performed using the BRUKER “zgesgp” pulse sequence (excitation sculpting with gradients). Additionally, the following acquisition parameters were used: 13.05 µs 90-degree pulse, 0.5 s relaxation delay, 64 K data points, 64 scans, 3.89 s acquisition time, and 10.03 ppm spectral width.

### 4.5. Spectral Processing and Metabolite Quantitation

Spectra were processed using the Chenomx NMR Suite Professional 7.7 software (Chenomx Inc., Edmonton, AB, Canada): phasing and baseline correction were performed, and the pH was calibrated using the resonances from imidazole. Spectra were referenced to DSS methyl peaks at 0.00 ppm. The same peak was also used as a chemical shape indicator, i.e., as an internal standard for quantitation. Thirty-nine metabolites were quantified in the ^1^D ^1^H NMR spectra of LT muscle extracts using the Profiler module of the Chenomx NMR Suite Professional software with an in-built 1D spectral library. Quantitation was based on comparing the area of selected metabolite peaks with the area under the DSS methyl peak, which corresponded to a known concentration of 0.5 mM in all samples. The resulting metabolite concentration tables (37 metabolites × 24 samples) were exported to Excel, where sample identifiers were added.

### 4.6. Statistical Data Analysis and Bioinformatics

Metabolomic data were analyzed using the open-source MetaboAnalyst 5.0 tool (http://www.metaboanalyst.ca/, accessed on 22 February 2022). Metabolite concentration tables were first uploaded to MetaboAnalyst, and data were log-transformed and Pareto-scaled before analysis. Supervised Partial Least Squares-Discriminant Analysis (PLS-DA) analyzes were performed with cross-validation using the LOOCV method and the performance measure “accuracy”. To investigate the production system and the growth rate effects on meat metabolomics, the following pairwise comparisons using the significance analysis of metabolites (SAM) approach were conducted to elucidate the specific differences between groups: FH versus PL, elucidating the effect of growth rate; and FL versus PH in regards to the feeding system. The more the variable deviates from the ‘‘observed-expected’’ line, the more likely it is to be significant. Colored dots represent features that exceed the specified threshold. The variable importance in the projection (VIP) was used to rank the metabolites based on their importance in discriminating the different groups. Metabolites with the highest VIP values (>1.0) are the most powerful group discriminators. 

Metabolomics data were further processed with bioinformatics by performing the pathway enrichment analyses using metabolite data sets for each group, according to Xia and Wishart [42]. The compound name was standardized according to KEGG ID, the pathway algorithms used were global test and relative-betweenness centrality, and the library chosen was *Bos taurus*.

## 5. Conclusions

The type of beef cattle production system and growth rate modulate the animal metabolism, changing meat metabolites composition and metabolic pathways, which impacts meat composition and quality development. This study provided great insight into differential meat metabolomics across animals finished on two divergent types of feeding systems and growth rates commonly used globally. Growth rate impacts meat metabolism through changes in energy metabolism and associated pathways, with FH animals presenting greater meat concentration of compounds correlated with energy production, such as succinate, AMP, inosine, G1P, and choline. Thus, the growth rate appears to be the main factor responsible for manipulating energy metabolism in fed cattle. The feeding system may be more responsible for altering protein and lipid metabolisms, with a depletion of protein regarding meeting the energy demand of grass-fed cattle. Despite the robustness of the model and accuracy of the data, further validation is needed using a high number of animals.

## Figures and Tables

**Figure 1 metabolites-12-00640-f001:**
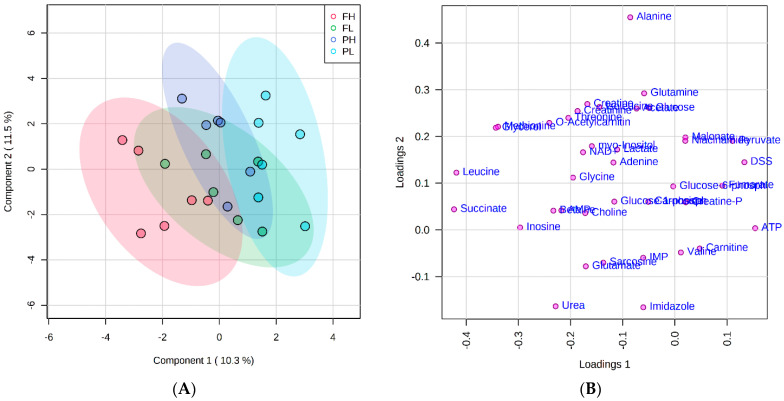
Partial least squares-discriminant analysis (PLS-DA) score plot (**A**) and loading plot (**B**) of metabolome distribution according to finishing system and rate of gain. FH: feedlot system with high average daily gain (ADG); (FL) feedlot system with low ADG; (PH) pasture system with high ADG; and (PL) pasture system with low ADG.

**Figure 2 metabolites-12-00640-f002:**
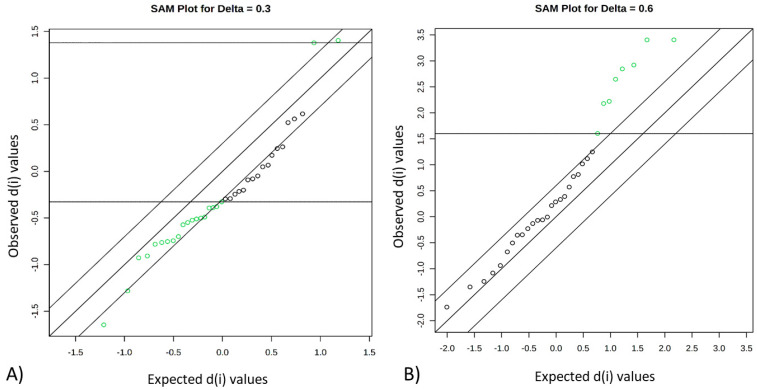
Significant features identified by significance analysis of metabolites (SAM). Green dots represent features that exceed the specified threshold, which represent differences in abundance among groups: FH versus PL (**A**) and FL versus PH (**B**). FH: feedlot system with high average daily gain (ADG); (FL) feedlot system with low ADG; (PH) pasture system with high ADG; and (PL) pasture system with low ADG.

**Figure 3 metabolites-12-00640-f003:**
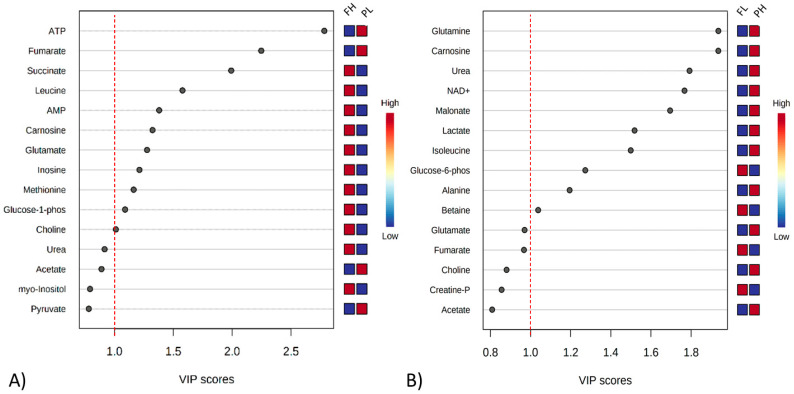
Variable importance in projection (VIP) plot value obtained from meat extracts classified as FH and PL (**A**), and FL and PH (**B**). VIP cut-off of around 1.0. FH: feedlot system with high average daily gain (ADG); (FL) feedlot system with low ADG; (PH) pasture system with high ADG; and (PL) pasture system with low ADG.

**Figure 4 metabolites-12-00640-f004:**
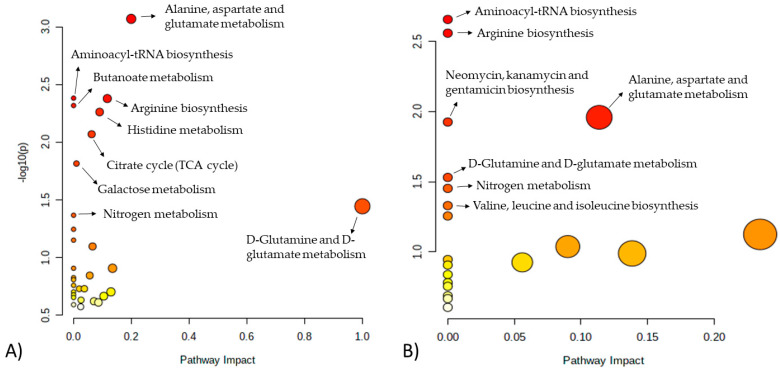
Metabolomics pathways of the meat extract revealed significant differences in the drawn pathway according to (**A**) FH versus PL and (**B**) FL versus PH. In the scatter plot, the x-axis indicates the impact on the pathway, whereas the y-axis indicates significant changes in the pathway by detected metabolites. Darker nodes represent higher *p*-values from the enrichment analysis, while larger nodes reflect greater impact from the pathway topology analysis. FH: feedlot system with high average daily gain (ADG); FL: feedlot system with low ADG; PH: pasture system with high ADG; and PL: pasture system with low ADG.

**Table 1 metabolites-12-00640-t001:** Descriptive analysis of metabolites concentrations (mg/g of fresh meat).

Metabolites	Minimum	Maximum	Mean	Std Error
AMP	0.01	0.05	0.03	0.002
ATP	0.00	0.01	0.00	<0.001
Acetate	0.01	0.03	0.01	<0.001
Adenine	0.02	0.05	0.04	0.001
Alanine	0.17	0.42	0.31	0.011
Betaine	0.06	0.20	0.11	0.004
Carnitine	0.27	0.67	0.45	0.016
Carnosine	0.36	2.73	1.88	0.085
Choline	0.02	0.05	0.031	0.001
Creatine	2.57	5.53	4.64	0.094
Creatine phosphate	0.02	0.20	0.05	0.005
Creatinine	0.01	0.05	0.03	0.001
Fumarate	0.00	0.01	0.00	<0.001
Glucose	0.17	0.52	0.37	0.014
Glucose-1-phosphate	0.05	0.11	0.07	0.003
Glucose-6-phosphate	0.08	0.67	0.43	0.020
Glutamate	0.02	0.05	0.03	0.001
Glutamine	0.12	0.48	0.26	0.014
Glycerol	0.16	0.49	0.27	0.017
Glycine	0.05	0.15	0.09	0.003
IMP	0.06	0.44	0.31	0.012
Inosine	0.01	0.05	0.02	0.001
Isoleucine	0.01	0.03	0.02	<0.001
Lactate	38.01	97.09	7.66	0.295
Leucine	0.01	0.04	0.03	0.001
Malonate	0.10	0.21	0.14	0.005
Methionine	0.01	0.04	0.02	0.001
Myo-Inositol	0.03	0.09	0.05	<0.001
NAD+	0.01	0.04	0.02	0.001
Niacinamide	0.01	0.03	0.02	0.007
O-Acetylcarnitine	0.15	0.32	0.23	<0.001
Pyruvate	0.00	0.02	0.01	0.002
Sarcosine	0.01	0.03	0.02	0.005
Succinate	0.04	0.17	0.11	0.008
Threonine	0.08	0.23	0.16	0.095
Urea	0.24	52.64	10.56	0.001
Valine	0.02	0.05	0.03	0.003

**Table 2 metabolites-12-00640-t002:** Significant features representing differences among the groups identified by significance analysis of metabolites (SAM). Delta = 0.3 for FH versus PL; and Delta = 0.6 for FL versus PH comparisons.

Metabolites	D.Value	Stdev	Raw *p*	Q.Value
FH versus PL				
Leucine	−1.64	0.13	<0.01	0.08
Fumarate	1.40	0.38	0.01	0.11
ATP	1.38	0.54	0.01	0.11
Succinate	−1.28	0.36	0.02	0.13
FL versus PH				
Glutamine	3.40	0.41	<0.01	0.04
Carnosine	3.43	0.41	<0.01	0.04
Urea	2.92	0.44	0.01	0.10
NAD+	2.84	0.45	0.01	0.10
Malonate	2.65	0.46	0.02	0.12
Lactate	2.22	0.49	0.04	0.19
Isoleucine	2.18	0.50	0.04	0.19

**Table 3 metabolites-12-00640-t003:** Results from polar meat extract pathway analysis from comparisons between FH and PL and FL and PH cattle.

Pathway Name	TC	Hits	Raw *p*	-log10 (*p*)	Holm *p*	FDR	Impact
FH versus PL							
Alanine, aspartate and glutamate metabolism	28	3	<0.001	3070	0.071	0.071	0.199
Aminoacyl-tRNA biosynthesis	48	3	0.004	2382	0.344	0.091	0
Arginine biosynthesis	14	2	0.004	2379	0.344	0.091	0.116
Butanoate metabolism	15	2	0.004	2318	0.388	0.091	0
Histidine metabolism	16	2	0.005	2262	0.437	0.091	0.090
Citrate cycle (TCA cycle)	20	2	0.008	2069	0.672	0.119	0.062
Galactose metabolism	27	2	0.015	1815	1	0.183	0.009
D-Glutamine and D-glutamate metabolism	5	1	0.035	1444	1	0.376	1
Nitrogen metabolism	6	1	0.042	1367	1	0.400	0
FL versus PH							
Aminoacyl-tRNA biosynthesis	48	3	0.002	2656	0.185	0.116	0
Arginine biosynthesis	14	2	0.002	2558	0.229	0.116	0
Alanine, aspartate and glutamate metabolism	28	2	0.010	1959	0.901	0.249	0.113
Neomycin, kanamycin and gentamicin biosynthesis	2	1	0.011	1925	0.962	0.249	0
D-Glutamine and D-glutamate metabolism	5	1	0.029	1530	1	0.493	0
Nitrogen metabolism	6	1	0.035	1452	1	0.493	0
Valine, leucine and isoleucine biosynthesis	8	1	0.046	1330	1	0.560	0

TC: the total number of compounds in the pathway; Hits: the actual matched number from the user uploaded data; Raw *p*: the original *p*-value calculated from the enrichment analysis; Impact: the pathway impact value calculated from pathway topology analysis.

## Data Availability

Data is contained within the article.

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
