# Peer review of "Impact of Cattle Feeding Strategy on the Beef Metabolome"

_metabolites, 2022, doi:10.3390/metabo12070640_

Round 1
Reviewer 1 Report
The authors tried to explain differences in meat quality, when animals are reared and fed in intensive or extensive systems, through the use of metabolomics. The study seems to be innovative and the results are clearly expressed. The discussion on the individuation of metabolities that change meat quality is well presented even if it need further investigations.
Author Response
Thank you for your kind comments. We fully agree that these data are quite useful and help us understand more fully changes in muscle that occur during different management strategies across the globe.

Reviewer 2 Report
There is repetitive use of common reference [7]. Hence, change or increase diversity of references.
Author Response
See comment to editor. Again, the present data are part of a large line of inquiry we started several years ago. The first results, more traditional in nature were published by Gomez et al. (2022), which is exactly why we repeatedly cite the Gomez paper. Observations in that paper and those cattle are exactly what we are trying to explain using this technology.

Reviewer 3 Report
The authors analyzed the impact of different feeding systems on metabolites, mainly from bovine muscle.
The comments are general, however the authors need to make a number of major changes to be considered for publication.
Line 34- In what sense.
Table 1- Reduce by 2 decimal places and homogenize.
Figure 1- not clearly seen.
Figure 2- not clearly seen.
Table 2-reduce 2 decimals and homogenize.
The discussion is very poor.
Line 236- Performed for all 36 samples.
Conclusion: Gives more emphasis to the results.
The literature review is poor.
Author Response
Comment - Line 34- In what sense.
Answer: Additional information was added to the conclusion section
Comment - Table 1- Reduce by 2 decimal places and homogenize.
Answer: Corrected
Comment - Figure 1- not clearly seen.
Answer: Figure quality has been improved
Comment - Figure 2- not clearly seen.
Answer: Figure quality has been improved
Comment - Table 2-reduce 2 decimals and homogenize.
Answer: Corrected
Comment - The discussion is very poor.
Answer: The discussion has been improved
Comment - Line 236- Performed for all 36 samples.
Answer: Metabolomic analyses was performed for 24 samples, it was added to the section 4.2
Comment - Conclusion: Gives more emphasis to the results.
Answer: Modified accordingly
Comment - The literature review is poor.
Answer: More references have been included in the text

Reviewer 4 Report
The present study explored changes in the beef metabolome of animals subjected to differing finishing systems and growth rates using an NMR-based method. Many metabolic biomarkers and possible metabolic pathways were found. The findings contribute to a better understanding of the influence of the beef cattle production system and growth rate on animal metabolism. However, the current version is not suitable for publication in Metabolites.
Major concerns
1. Why the author compare FH group with PL group, FL group with PH? It seems that the author aimed to identify metabolites and metabolic pathways related to growth rate by comparing FH with the PL group. However, the FH and PL groups are different in two factors: finishing system and growth rate. So how does the author determine whether the changes in metabolites and metabolic pathways are caused by the finishing system, growth rate, or both? Maybe FH vs. FL or PH vs. PL is much better?
2. The present study identified only 37 compounds in beef using an NMR-based method. The number of detected metabolites is relatively small. LC-MS and GC-MS-based metabolomics have been widely used in human, animal, and plant biological studies. Compared with NMR, LC-MS and GC-MS-based metabolomics can detect a relatively large number of metabolites. Why did the author choose NMR-based metabolomics but not LC-MS and GC-MS-based metabolomics?
3. The sample size for the study was n=6 per group. This sample size is too small for metabolomics study in animals. The recommended sample size is n>8.
Minor concerns
1. Figure 1A—Please indicate what are F-H, F-L, P-H and P-L in the figure citation.
2. Figure 1B—Avoid overlap of the text.
3. In the discussion, the author said greater ATP may lead to higher concentrations of AMP sooner. However, the present study found that ATP was greater in beef from PL group, while AMP was greater in that of FH cattle. It seems that ATP concentrations is negatively correlated with AMP concentration.
4. Figure 3—Please add figure legend for the color and node size.
5. Data availability – there are a number of repositories for metabolomic data and the data should be made available.
Author Response
Comment - The present study explored changes in the beef metabolome of animals subjected to differing finishing systems and growth rates using an NMR-based method. Many metabolic biomarkers and possible metabolic pathways were found. The findings contribute to a better understanding of the influence of the beef cattle production system and growth rate on animal metabolism. However, the current version is not suitable for publication in Metabolites.
Answer: Thank you for your comments. We considered each comment very carefully and modified the manuscript accordingly, hoping to clarify ambiguous points.
Major concerns
Comment - Why the author compare FH group with PL group, FL group with PH? It seems that the author aimed to identify metabolites and metabolic pathways related to growth rate by comparing FH with the PL group. However, the FH and PL groups are different in two factors: finishing system and growth rate. So how does the author determine whether the changes in metabolites and metabolic pathways are caused by the finishing system, growth rate, or both? Maybe FH vs. FL or PH vs. PL is much better?
Answer: We understand your concerns but this work is part of a larger experiment, so the pairwise comparisons presented are consistent with earlier investigations and phenotypic data presented by Gomez et al., (2022). We have tried to clear this up in the revision to avoid any ambiguity and further referenced the earlier work. The decision for considering FH vs PL, and FL vs PH is based on the phenotypic responses that were identified in the large experiment to further our understanding at the metabolome level. Moreover, since the animals belonging to the groups FH vs PL presented a greater difference on growth rate, the metabolomics profiling is a perfect way to decipher the metabolism differences and any shifts that might exist between high and low growth rates. While FL and PH did not present differences on performance parameters, animals respond very similarly to the treatments, in this case, metabolomics differences on their metabolism could be a consequence of differences in the feeding system. The same strategy was applied in another large study described by Antonelo et al., (2022) and we cite this manuscript to further clarify this specific point.
Antonelo, D.S.; Gómez, J.F.M.; Silva, S.L.; Beline, M.; Zhang, X.; Wang, Y.; Pavan, B.; Koulicoff, L.A.; Rosa, A.F.; Goulart, R.S.; et al. Proteome Basis for the Biological Variations in Color and Tenderness of Longissimus Thoracis Muscle from Beef Cattle Differing in Growth Rate and Feeding Regime. Food Res. Int. 2022, 153, doi:10.1016/j.foodres.2022.110947.
Gómez, J.F.M.; Antonelo, D.S.; Beline, M.; Pavan, B.; Bambil, D.B.; Fantinato-Neto, P.; Saran-Netto, A.; Leme, P.R.; Goulart, R.S.; Gerrard, D.E.; et al. Feeding Strategies Impact Animal Growth and Beef Color and Tenderness. Meat Sci. 2022, 183, doi:10.1016/j.meatsci.2021.108599.
Comment - The present study identified only 37 compounds in beef using an NMR-based method. The number of detected metabolites is relatively small. LC-MS and GC-MS-based metabolomics have been widely used in human, animal, and plant biological studies. Compared with NMR, LC-MS and GC-MS-based metabolomics can detect a relatively large number of metabolites. Why did the author choose NMR-based metabolomics but not LC-MS and GC-MS-based metabolomics?
Answer: We fully understand the opinion/suggestion of the referee about using other high-throughput methods based on mass spectrometry; however, it is important to emphasize that NMR is also widely used for metabolomics studies and considered as a reference method. Each of these methods has their pros and cons and there is no unique best method. The choice of NMR in our case is mainly because of the ease of sample preparation for both quantitative and qualitative evaluation of a wide range of metabolites in a complex and heterogeneous matrix (in our case meat). Thus, this approach is able to offer a clear identification of samples, in our case, meat profiles, resulting in less biochemical modification due to manipulation. Using LC-MS and GC-MS methodologies we could have the fingerprint or a target analysis with some (dozens or hundred) chosen metabolites. However, in our case, the metabolomics profile obtained from NMR, should bring us a better understanding about this data, since we were not looking for some specific metabolites or pathways, in the case of the target analysis. Also we do not believe that the fingerprint will be enough in this case to explain specific differences between treatments, since we would like to see whatever the differences were between treatments about the metabolomics profile.
In addition, as well explained by Ocampos et al. (2021), NMR and LC-MS the most widely used techniques for the investigation of natural products, with complementary characteristics. NMR is considered reliable and reproducible tool. By analyzing a combination of obtained spectra and correlation maps, one can obtain chemical structures unambiguously. It is useful even in mixtures, giving the relative concentration of all the compounds in the sample, and when the correct parameters are used, it can also give the absolute concentration of each compound in a given sample in line to our objectives in this trial.
Ocampos FMM, de Souza AJB, Antar GM, Wouters FC, Colnago LA. Phytotoxicity of Schiekia timida Seed Extracts, a Mixture of Phenylphenalenones. Molecules. 2021; 26(14):4197. https://doi.org/10.3390/molecules26144197
Comment - The sample size for the study was n=6 per group. This sample size is too small for metabolomics study in animals. The recommended sample size is n>8.
Answer: We respected and understand your regarding this issue, however, the n of this study was 8 animals per group, however due to circumstances beyond our control, animals were eliminated from the study. Even so, we believe our data are accurate because they fit well with the phenotypic observations, as previously reported. Furthermore, a power analysis was conducted and confirmed the robustness of the model and accuracy of the data. In addition, this is an exploratory study in the field and this is something that is now added in the conclusions that further validation is needed using high number of animals.
Minor concerns
Comment - Figure 1A—Please indicate what are F-H, F-L, P-H and P-L in the figure citation.
Answer: Corrected
Comment - Figure 1B—Avoid overlap of the text.
Answer: We followed the guidelines that each figure should be understood without referring to the text. We think that the overlap is minor and the caption can be kept in its present form.
Comment - In the discussion, the author said greater ATP may lead to higher concentrations of AMP sooner. However, the present study found that ATP was greater in beef from PL group, while AMP was greater in that of FH cattle. It seems that ATP concentrations is negatively correlated with AMP concentration.
Answer: This was a result of some confusion among authors. Corrected
Comment - Figure 3—Please add figure legend for the color and node size.
Answer: Corrected
Comment - Data availability – there are a number of repositories for metabolomic data and the data should be made available.
Answer: We thank the reviewer for her/his advice. The data will be made available on request. A data paper is under preparation to make the data available in a better way.

Round 2
Reviewer 3 Report
The article should be accepted for publication as the authors considered all the revisions, but the following should be urgently corrected:
Homogenise the values in table 1. Decimals should be reduced by the standard error value, as was done for the above ....
Correct figure 1. A subtitle was cut off.
Reduce the decimals in table 2, so that all tables present values with the same number of decimals.
Author Response
Reviewer 3 – Round 2
The article should be accepted for publication as the authors considered all the revisions, but the following should be urgently corrected:
Homogenise the values in table 1. Decimals should be reduced by the standard error value, as was done for the above ....
Corrected
Correct figure 1. A subtitle was cut off.
Corrected
Reduce the decimals in table 2, so that all tables present values with the same number of decimals.
Corrected

Reviewer 4 Report
The authors have addressed all my concerns. I recommend accepting the paper in its present form.
Author Response
We affectionately thank the reviewer for the comments and corrections
